# Comment on Subhadra et al. Significant Broad-Spectrum Antiviral Activity of Bi121 against Different Variants of SARS-CoV-2. *Viruses* 2023, *15*, 1299

**DOI:** 10.3390/v15112268

**Published:** 2023-11-17

**Authors:** Žarko Kulić

**Affiliations:** Preclinical Research and Development, Willmar Schwabe GmbH & Co. KG, Willmar-Schwabe-Straße 4, D-76227 Karlsruhe, Germany; zarko.kulic@schwabe.de; Tel.: +49-721-4005-9669

The article “Significant Broad-Spectrum Antiviral Activity of Bi121 against Different Variants of SARS-CoV-2” by Subhadra et al. [1] presents a standardized polyphenolic-rich extract from *Pelargonium sidoides* called Bi121, with data on its antiviral activity against different SARS-CoV-2 variants and the underlying fractions of Bi121 responsible for this activity. The authors narrow down the activity to one specific compound called Neoilludin B and show RNA-intercalating activity of the compound in RNA viruses by an in silico structural modelling. Due to the omnipresent medical need in the recent past, we highly appreciate research and development of therapeutics against SARS-CoV-2 in general and respective antiviral phytopharmaceutical drugs in particular. However, we have severe concerns about certain aspects presented in the aforementioned study and therefore want to draw the attention of the journal editors, reviewers, and readers of the article towards these aspects.

One major prerequisite for valid scientific studies is reproducibility. To facilitate reproducibility, a comprehensive characterization of the study material is crucial. While working in the field of pharmacology and medical sciences, studied drugs need to be sufficiently characterized. If the used drug is a single compound with a known chemical structure, an unambiguous name and/or structure of the molecule needs to be provided, with the specification of the purity of the substance, e.g., 98% (m/m) acetylsalicylic acid. Working with plant extracts as active principles, on the other hand, makes this characterization of the studied drug more complicated since plant extracts are multicomponent mixtures, with thousands of individual compounds in different concentrations. This is further complicated by seasonal and geographical variabilities of the plant material used for extraction. Thus, concentrations of the compounds in a plant vary from season to season due to variations, e.g., in precipitation and other environmental factors. Yet another factor is the process used for extraction of the plant material. The use of different solvents, extraction times, temperatures, pressures, etc., will affect the concentrations of the different compounds in the final extract. Also, the plant material itself needs to be properly authenticated since some plants within a plant family look quite similar and the possibility of adulteration and/or false authentication needs to be ruled out. Due to this multidimensionality of plant extracts as active substances, the characterization of the used material should at least include a deposit of a voucher specimen, the thorough description of the extraction process, and chemical fingerprints by methods that are suitable for the assessment of substance mixtures, like high performance liquid chromatography (HPLC), thin layer chromatography (TLC), nuclear magnetic resonance (NMR), and the like. A consensus statement on the best practice in characterizing multicomponent mixtures like plant extracts was recently published, covering all these aspects in deep detail [2].

In the study of Subhadra et al. [1], the *Pelargonium sidoides* extract Bi121 was prepared following a protocol described in literature [3], with some modifications like increased extraction temperature and the use of ultrasonication. Although described as a “standardized extract”, the standardization parameter was not specified anywhere in the article. Common standardization parameters may include amount ranges of characteristic or active compounds within the extract, e.g., 4–6% (m/m) of compound X. Examples of standardization specifications of phytopharmaceuticals can be found in many pharmacopoeias, e.g., in the European pharmacopoeia (Ph. Eur.). While the extraction process for Bi121 is described to a sufficient extent, the plant material used for said extraction was not authenticated, and no voucher specimens were deposited by the authors. As a chemical fingerprint of Bi121, an HPLC chromatogram was presented (see [1], Figure 5), although no detection wavelength is specified for the chromatogram, which is crucial information. The peaks within this chromatogram are annotated by incremented numbers from 1–15 and represent “characteristic compound peaks” according to the figure caption. However, the assignment of 1–15 to “characteristic compounds” cannot be found anywhere in the article. Even more striking, the chromatographic profile of Bi121 seems to be completely identical to the chromatographic profile of the *Pelargonium sidoides* extract EPs^®^ 7630 published previously in another MDPI journal [4]: Please compare Figure 5 from Subhadra et al. [1] to Figure 1 from Roth et al. [4]. Although, in theory, the phytochemical compositions of Bi121 and EPs^®^ 7630 and thus also chromatographic profiles could be so similar that they become indistinguishable, this theoretical assumption is very unlikely due to the following reasons:
1.It is very unlikely that the respective plant material used for preparation of the analyzed extract batches Bi121 and EPs^®^ 7630 is identical. At least some seasonal variability is to be expected.2.The extraction processes of Bi121 and EPs^®^ 7630 are different. Bi121 is prepared by extraction of the plant material with pure water using ultrasonication followed by an adsorption to polyvinylpyrrolidone with a subsequent elution with NaOH to enrich the polyphenols. In contrast, EPs^®^ 7630 is extracted with an aqueous 11% (m/m) ethanol solution with deviating further processing steps. The different extraction solvents and the different subsequent processing should yield different phytochemical compositions of the two extracts and thus differences in HPLC profiles.3.The most striking argument is that the described HPLC methods used to generate the HPLC chromatograms in [1,4], respectively, use different HPLC instruments, different separation columns, and different eluents. In practice, this difference in methodology will most likely yield different chromatographic profiles even if the very same plant extract is used.


From a scientific point of view, the fact that the HPLC profile presented by Subhadra et al. is identical to an HPLC profile formerly published by another independent research group appears to be startling, and it is necessary to exclude any implication of plagiarism of the figure in order to be in accordance with prevailing scientific publication standards.

Apart from the characterization of the extract Bi121, one further point of discussion is the putative identification of Neoilludin B as an active substance. Illudins were found only in mushrooms so far, and literature on this compound class is scarce. While finding new and unexpected compounds in plants is a common occurrence for natural product researchers, a finding that a (so far) mushroom-specific compound is found in plants must be supported by a higher evidence level, like isolation of the compound followed by structure verification by NMR. The identification by Subhadra et al. using compound library comparison based on only mass spectrometric data is very putative since isomeric substances cannot be excluded, and any other substance having the same sum formula may be the active substance. Furthermore, even if the presence of Neoilludin B is confirmed, it must be excluded that the substance was introduced by an adulteration of the plant material with (parasitic) mushrooms, which leads back to the need for proper authentication of the plant material. A higher evidence level for the identification of the substance is required for the study in particular since the active principle of Neoilludin B is further underlined by an in silico target prediction and validation. A potential false identification of the active substance would render these in silico studies useless.

Considering all the aforementioned arguments, concerns are raised on what conclusions may be drawn from this study, where the standardization of the studied drug is not specified, the studied drug is not characterized by the assignment of “characteristic compounds” to the presented annotated HPLC peaks, the HPLC profile itself is at least suspicious of plagiarism, and the postulation of the active principle is based on a very putative identification of a single compound. These concerns render the validity of the study very limited and call for a more comprehensive phytochemical reassessment of the studied material.

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
