# Peer review of "Comment on Subhadra et al. Significant Broad-Spectrum Antiviral Activity of Bi121 against Different Variants of SARS-CoV-2. Viruses 2023, 15, 1299"

_viruses, 2023, doi:10.3390/v15112268_

Round 1

Reviewer 1 Report

Comments and Suggestions for Authors

The manuscript is a letter to the editor that critically examines the findings of Subhadra et al.'s study. The primary concern exposed by the author is the identical HPLC profiles of two extracts, and second the identification of Neoilludin B in plants. The letter's main contribution is its emphasis on the importance of scientific rigor and authenticity in research publications.

Major Comments:

Identical HPLC Profiles (Lines 52-76):

Concern: The HPLC profile of Bi121 appears to be identical to that of the Pelargonium sidoides extract EPs® 7630 from another study. Given the different methodologies used, this similarity is highly suspicious.

Recommendation: I agree with the author's letter. The authors of the original study should provide a clear explanation for the identical profiles. If necessary, they should reproduce their results to confirm their findings. The validation that both figures are indeed similar lends credence to this concern. [Reference: Roth et al. [4]]

Identification of Neoilludin B (Lines 77-91):

Concern: The identification of Neoilludin B in plants, based solely on mass spectrometric data, is questionable. The possibility of contamination from parasitic mushrooms should also be explored.

Recommendation: I agree. The original authors should provide more comprehensive evidence, such as NMR data, to support the identification. Ensuring proper authentication of the plant material is crucial.

Overall Recommendation: Given the validation that the figures from the two studies are indeed similar, it's crucial for the authors of the original study to address the concerns raised in this letter comprehensively. Providing clear explanations, evidence, and, if necessary, reproducing the results will be essential to ensure the study's credibility and authenticity.

Author Response

Thank you for the assessment of my comment!

Since your concerns and recommendations are in agreement with my comment/letter to the editor, I assume that no further revisions are needed from my side.

Apart from your concern, I have added a Conflict of Interest statement (marked in the resubmission), which I forgot in my first manuscript.

If additional revisions are needed from my side, please let me know! Otherwise, I thank you for a recommendation for publishing my comment.

Reviewer 2 Report

Comments and Suggestions for Authors

Zarko Kulic has provided certain important points about the original manuscript by Subhadra et al.,

Major concern:

In the comments, the author have pointed out a potential plagiarism, that needs to be fixed. 

Minor concern: 

The author mention about describing the details on chemical structures. While it is vital to include chemical structures for novel chemical entities, including structures for known (established chemicals) is helpful but not mendatory. 

Author Response

Thank you for the assessment of my comment!

Since your major concern is about the original study’s potential plagiarism I pointed out in the comment, this concern is not about my comment and thus, does not need to be answered by myself.

Your minor concern is about my comment. I agree with you, that including structures for known compounds is helpful but not mandatory. However, I did not asked for details about the chemical structure of Neoilludin B, which is already known. Instead, I raised a concern, that said compound is mushroom specific as described by literature so far, and yet the authors claim to have found it in their plant extract by LC-MS, which is not an unambiguous method for compound verification. My concern is, that if an unexpected compound is found, it needs strong evidence by verification with an unambiguous method (like NMR). Furthermore, it needs to be verified that the compound is not introduced by (parasitic) mushrooms present in the plant material used for the study.

As a comparison: if a gene or a protein, which is described in literature to be virus specific so far (e.g. RNA replicase) is found by a study to be present in a bacterium, this finding has to be supported by unambiguous characterization, and it needs to be excluded, that the gene/protein was introduced by an infection of the bacterium with a virus.

I hope this explanation resolves your minor concern.

Apart from your concern, I have added a Conflict of Interest statement (marked in the resubmission), which I forgot in my first manuscript.

I hope that my explanations and revision are comprehensive to meet your requirements for the recommendation for publishing my comment.